# Preparation of MoS_2_@PDA-Modified Polyimide Films with High Mechanical Performance and Improved Electrical Insulation

**DOI:** 10.3390/polym16040546

**Published:** 2024-02-17

**Authors:** Xian Cheng, Chenxi Wang, Shuo Chen, Leyuan Zhang, Zihao Liu, Wenhao Zhang

**Affiliations:** 1School of Electrical and Information Engineering, Zhengzhou University, Zhengzhou 450001, China; chengxian@zzu.edu.cn (X.C.); wangchenxi@gs.zzu.edu.cn (C.W.); zly19991124@163.com (L.Z.); 15139085273@163.com (Z.L.); zwh1823641629@163.com (W.Z.); 2He’nan Engineering Research Center of Power Transmission and Distribution Equipment and Electrical Insulation, Zhengzhou 450001, China

**Keywords:** PI, MoS_2_ nanosheets, composite, mechanical properties, aging life

## Abstract

Polyimide (PI) has been widely used in cable insulation, thermal insulation, wind power protection, and other fields due to its high chemical stability and excellent electrical insulation and mechanical properties. In this research, a modified PI composite film (MoS_2_@PDA/PI) was obtained by using polydopamine (PDA)-coated molybdenum disulfide (MoS_2_) as a filler. The low interlayer friction characteristics and high elastic modulus of MoS_2_ provide a theoretical basis for enhancing the flexible mechanical properties of the PI matrix. The formation of a cross-linking structure between a large number of active sites on the surface of the PDA and the PI molecular chain can effectively enhance the breakdown field strength of the film. Consequently, the tensile strength of the final sample MoS_2_@PDA/PI film increased by 44.7% in comparison with pure PI film, and the breakdown voltage strength reached 1.23 times that of the original film. It can be seen that the strategy of utilizing two-dimensional (2D) MoS_2_@PDA nanosheets filled with PI provides a new modification idea to enhance the mechanical and electrical insulation properties of PI films.

## 1. Introduction

Polyimide (PI) is widely used in the field of electronic devices, aerospace and high-performance coatings due to its superior thermal stability, mechanical strength, and chemical resistance [1,2,3]. In addition, PI materials are suitable for high-voltage insulation due to their superior electrical insulation properties, especially in the field of turn-to-turn insulation where they play a vital role [4,5]. As a key component of electrical equipment such as motors and transformers, inter-turn insulation needs strong electrical insulation properties to cope with high amplitude and frequency pulses of high voltage. More critically, the inter-turn insulation demands superior mechanical and frictional properties to reduce air gaps formed during the winding process due to the poor fit of insulating materials. These air gaps can lead to partial discharges under the action of strong electric fields. Moreover, strong leakage in electromagnetic fields may cause mechanical damage to the winding ends. The above factors are the main reasons for the deterioration of the insulation properties of PI materials [6,7,8].

Two-dimensional transition metal sulfides have received widespread attention in scientific research and industry due to their unique physical and chemical properties [9,10,11]. Molybdenum disulfide (MoS_2_) is a two-dimensional material with a special layered structure and a high degree of surface activity, which makes it excellent in terms of mechanical and friction properties [12,13]. Yuan et al. [14] found that doping a small amount of MoS_2_ (0.75 wt.%) can significantly enhance the mechanical properties of MoS_2_/PI nanocomposite films. Liu et al. [15] found that MoS_2_ can effectively reduce the deformation of the PI wear surface, thus reducing and stabilizing friction. Guo et al. [16] doped phenolic resin (CPF) composites with 10 wt.% MoS_2_, which resulted in a 56% decrease in wear rate compared to pure CPF. The above studies show that MoS_2_-modified polymer composites have great strength and toughness, a low coefficient of friction, and high wear resistance. However, since MoS_2_ is a semiconductor material with a narrow forbidden bandwidth, its application in the field of high-voltage insulation is limited. Pang et al. [17] showed that in terms of pressure cylinders in high-voltage switchgear, excessive accumulation of MoS_2_ reduces the arc extinguishing and breaking capacity of the switchgear, which leads to a decrease in overall insulation performance. Meanwhile, the high surface energy of nanomaterials tends to cause agglomeration, so the interfacial interaction between MoS_2_ and the PI matrix may also be affected by its dispersion [18,19,20].

Polydopamine (PDA) is a substance formed by the self-polymerization of dopamine in an alkaline environment, with a unique structure and functional groups. PDA has a great property boost effect on the mechanical properties of composites due to its ability to form various types of bonds with other materials, such as hydrogen and coordination bonds [21,22,23]. Dong et al. [24] found that PI composites doped with 5 wt.% of PDA-encapsulated graphitic carbon nitride had the highest breakdown field strength of 300 kV/mm, which was 67.6% higher than that of pure PI. Feng et al. [25] found that the interfacial interaction between filler and polyarylene ether nitrile was enhanced by a core–shell structural composite (MoS_2_@PDA). Sanusi Hamat et al. [26] found that a coupling agent (PDA) could increase the tensile Young’s modulus, flexural Young’s modulus, and compressive stress of kenaf fibers and the PLA matrix by 13.4%, 15.3%, and 30%, respectively. Therefore, the use of MoS_2_@ PDA core–shell structure nanoparticles doped with modified PI materials is expected to further enhance the mechanical and friction properties of the composites, as well as effectively improve the electrical insulation properties of the composites. This comprehensive performance improvement is essential to safeguard the insulation life of the material during long-term use and is also significant in improving the operational reliability of electrical equipment.

In this research, a core–shell structure of MoS_2_ encapsulated in PDA was strategically employed for the optimization and modification of PI composites. The PDA not only enhances the dispersion and compatibility of MoS_2_ within the PI matrix but also significantly increases the flexibility of the composite film. This crucial enhancement in flexibility is essential for the minimization of the partial discharge phenomenon, particularly those discharges caused by air gaps between coil turns. Moreover, the PDA layer acts as an effective barrier, markedly reducing the electronic excitation and migration in the MoS_2_ semiconductor. This substantial reduction is pivotal in significantly improving the overall electrical insulation properties of the composite material.

## 2. Materials and Methods

### 2.1. Materials

Thiourea (CH_4_N_2_S), ammonium molybdate tetrahydrate ((NH_4_)6Mo_7_O_24_·4H_2_O), Tris(hydroxymethyl) aminomethane (Tris, C_4_H_11_NO_3_), dopamine hydrochloride (C_8_H_11_NO_2_·HCl), and N,N-Dimethylacetamide (CH_3_CON(CH_3_)_2_) were purchased from Shanghai Macklin Biochemical Technology Co., Ltd. (Shanghai, China). The PI precursor (PAA) was purchased from Changzhou Runchuan Plastic Material Co., Ltd. (Changzhou, China) as well as the deionized water.

### 2.2. Synthesis of MoS_2_ Nanosheets

In this procedure, 0.144 g of ammonium molybdate tetrahydrate and 1.010 g of thiourea were accurately measured and introduced into a beaker containing 60 mL of water as the chosen solvent. The mixture was then stirred continuously for 2 h to ensure the sample was completely dissolved. Once fully dissolved, the homogenous solution was carefully transferred into the reactor. The reactor was subsequently heated for a duration of 12 h at a stable temperature of 180 °C. After the heating process, MoS_2_ was successfully obtained through a methodical process of filtration and subsequent drying under controlled conditions.

### 2.3. Synthesis of MoS_2_@PDA Nanosheets

Approximately 0.5 g MoS_2_ was dispersed in 250 mL of deionized water, ultrasonicated, and stirred for 0.5 h to obtain suspension A. Approximately 0.67 g of Tris was added to A, and we continued to stir it ultrasonically for 0.5 h to obtain suspension B. Finally, 0.1 g of dopamine hydrochloride was added to solution B, and it was placed on a magnetic stirrer at ambient temperature and stirred for 24 h. Then, it was filtered and dried in a vacuum drying oven for 12 h to obtain MoS_2_@PDA nanosheets.

### 2.4. Synthesis of MoS_2_/PI and MoS_2_@PDA/PI Films

Figure 1 shows the synthesis process for MoS_2_/PI and MoS_2_@PDA/PI films. Approximately 150 mg MoS_2_ was sonicated and dispersed in 5 mL N,N-dimethylacetamide and stirred for 30 min; then, 25 g of PAA was added and stirred for 24 h until the MoS_2_ nanosheets were evenly dispersed in the above solution. After eliminating the air bubbles using a vacuum pump, the solution was poured into a glass plate and coated with the film evenly. Finally, MoS_2_/PI film was obtained after heating the coating glass plate at different temperature gradients in an oven. The MoS_2_@PDA/PI composite films could be obtained by means of the above steps using the same content of MoS_2_@PDA nanosheets. In addition, the pure PI film was obtained by heating the PAA film. 

### 2.5. Materials Characterization

The microstructure and morphology of the as-prepared samples were characterized by field emission scanning electron microscopy (F-SEM; JEOL JSM-7800F, JEOL, Tokyo, Japan). The sample was dripped onto a copper grid and dried before observation under transmission electron microscopy (TEM), with the test temperature set at 30 °C and the magnification range from 5000 to 50,000 times. The lattice structure of MoS_2_ and MoS_2_@PDA composites was further characterized by TEM (JEOL JEM-2100), with the test temperature maintained at 30 °C and the magnification range from 12,000 to 120,000 times. The surface functional groups of the samples were analyzed using Fourier-transform infrared spectroscopy (FT-IR, Nicolet iS50, Nicolet, Green Bay, WI, USA), with the test temperature held at 30 °C and the wavelength range tested between 600 and 4000 cm^−1^. The crystalline structure of the samples was determined using powder X-ray diffraction (XRD, Rigaku Ultima IV with Cu-Kα radiation, λ = 0.15418 nm, Rigaku, Tokyo, Japan) over a range of 10–90° and X-ray photoelectron spectroscopy (XPS, AXIS Supra) over a range of 0–1200 eV, with the test temperature set at 30 °C. The friction and wear properties were measured using a friction wear test machine (MG2000). The experimental load was set to 120 N, the experimental duration was 5 min, and the speed of the friction wear testing machine was maintained at 200 r/min. The dielectric constant and loss tangent were obtained using a broadband dielectric spectrometer and impedance analyzer (Novocontrol Concept 80, Novocontrol Technologies, Frankfurt, Germany). The frequency range covered was from 10^2^ to 10^6^ Hz, and the test temperature was held constant at 30 °C. An electrode with a diameter of 30 mm was sprayed on both sides of the samples to facilitate the measurements.

## 3. Results

### 3.1. Morphology Analyses

The 2D nanostructure of MoS_2_ can be observed through the SEM and TEM images. In Figure 2a, the MoS_2_ nanosheets undergo a self-assembly process to form a nanoflower with a diameter of about 300 nm, and the size is relatively uniform. Figure 2b is the TEM image of a pure MoS_2_ nanosheet, which confirms the stacked structure [27]. In the highly magnified TEM image (Figure 2c), the color of the MoS_2_ single layer is lighter, indicating that the thickness of the nanosheet is thin [28]. The morphology and structure of the MoS_2_@PDA (Figure 2d) do not have larger changes compared with MoS_2_, and the thicker nanosheet layer (Figure 2e,f) illustrates the successful encapsulation of dopamine on the surface of MoS_2_. The PI film displays the smooth surface morphology in Figure 2g and benefits from the ultrasonic stirring of the nanosheets before the experiment, and MoS_2_ and MoS_2_@PDA exhibit a flat two-dimensional structure on the surface of the PI film. Compared with MoS_2_/PI film, the dispersion of MoS_2_@PDA/PI is better and more uniform (Figure 2h,i). In Figure 2j, the distribution of MoS_2_@PDA nanosheets can be clearly seen through the EDX-SEM mapping, and the concentrated area of Mo and S elements is consistent with the distribution of nanosheets in the SEM image.

### 3.2. Chemical Structure

XRD analysis can be used to obtain information about the structure of atoms or molecules and the composition of the sample. The XRD diffraction patterns of the five samples are shown in Figure 3a, MoS_2_ and MoS_2_@PDA had obvious diffraction peaks at 32.7°, 35.7°, 43.1°, and 57.7° which match with the (100), (102), (006), and (110) crystal planes of 2H-MoS_2_ nanosheets, respectively [10,29]. The PDA is an amorphous structure with no obvious diffraction peaks. In addition, the pure PI, MoS_2_/PI, and MoS_2_@PDA/PI films have a broad peak at around 20°, indicating the presence of partial crystallization in the amorphous PI which is consistent with other literature sources [30]. The successful preparation of composite films was confirmed by the diffraction peaks of MoS_2_/PI and MoS_2_@PDA/PI near the (100) and (110) crystal planes of MoS_2_. The faint diffraction peaks of MoS_2_/PI and MoS_2_@PDA/PI films are due to the low doping amount of the nanosheets. Figure 3b shows the FT-IR spectra of PI, MoS_2_/PI, and MoS_2_@PDA/PI. The peaks at 723 cm^−1^ could be ascribed to the C=O bending in PI, while those absorption peaks at 1718 and 1776 cm^−1^ could be assigned to the C=O of symmetrical stretching vibration and asymmetrical stretching vibration [31]. Due to the dehydration and cyclization of the PI precursor, there was no obvious characteristic absorption peak at about 1540 and 1645 cm^−1^. Instead, there was a stretching vibration of –C–N–C at 1376 cm^−1^, which indicated the amide to be replaced by the imide group [32,33]. From the above, it can be seen that imidization was almost complete, and this further proves the successful synthesis of the composite films in this research.

XPS characterization is a technique for analyzing the chemical valence states of elements on the surface of composite materials, and the survey spectra of MoS_2_@PDA/PI composite films can be seen in Figure 4a. As Figure 4b shows, the C 1s spectrum can be deconvoluted into three diffraction peaks which are located at 284.6, 285.6, and 287.1 eV assigned to C–C/C=C, C–N, and C–O bonds, respectively [34,35]. The N 1s fine spectrum of MoS_2_@PDA/PI exhibits an intense symmetrical peak centered at 399.6 eV, assigned to the imide (C–N) group. The N orbital shifts towards low energy compared to pure PI films in the literature, due to the non-covalent interaction between PI and MoS_2_@PDA [14]. In Figure 4d, the O 1s spectrum is deconvoluted into two energy levels, located at the center of 531.5 and 533.1 eV, corresponding to C=O and C–O, respectively [34].

### 3.3. Mechanical Properties

The tribological properties of pure PI, MoS_2_/PI, and MoS_2_@PDA/PI composite films are presented in Figure 5. For the pure PI film, the friction coefficient is 0.39 and the wear rate is 1.8 × 10^−4^ mm^3^/Nm. When MoS_2_ is incorporated, a decrease in both the friction coefficient and wear rate of the PI composite film is observed, and the values drop to 0.31 and 1.6 × 10^−4^ mm^3^/Nm, respectively. This reduction is attributed to the layered structure of MoS_2_, which features layers interconnected by weak van der Waals forces. These layers can easily slide over each other under shear forces. In terms of wear, MoS_2_ particles are found to migrate to the worn surface. They form a lubricating film, shifting wear from the composite material and its counterpart to the composite material. This shift effectively reduces the wear rate and friction coefficient of the composite. Furthermore, the excellent thermal conductivity of MoS_2_ assists in dissipating heat generated during friction. This dissipation prevents heat accumulation that could cause local softening and deformation of the material, thereby enhancing tribological performance [36].

The frictional properties are further improved by modifying MoS_2_ with PDA. The MoS_2_@PDA/PI composites have a friction coefficient of 0.24 and a wear rate of 1.3 × 10^−4^ mm^3^/Nm, which are significantly reduced by 38.4% and 27.8%, respectively, compared with the pure films. The incorporation of PDA provides active sites on the MoS_2_ nanosheets, improving their dispersion within the PI matrix and reducing aggregation. Therefore, the enhanced mechanical wear capacity of MoS_2_@PDA/PI can be attributed to the uniform dispersion of MoS_2_@PDA nanosheets in the PI matrix [37].

The tensile test results of pure PI, MoS_2_/PI, and MoS_2_@PDA/PI composite materials are shown in Figure 6. Among the three materials, MoS_2_@PDA/PI exhibits the best mechanical properties, followed by MoS_2_/PI in second place, and pure PI ranking as the least performant. Specifically, the tensile strength of the MoS_2_/PI composite film increases from 101 MPa to 124.6 MPa, the elastic modulus increases from 2.0 GPa to 2.4 GPa, and the elongation at breaks increases from 21.1% to 30.2%. Notably, with the introduction of PDA-modified MoS_2_, the tensile strength (146.2 MPa) of the composite material increases by 17.3% compared to MoS_2_/PI. Meanwhile, the elastic modulus (2.5 GPa) of MoS_2_@PDA/PI increases by 4.1%, and the elongation at break rises to 32.8% compared to MoS_2_/PI. This phenomenon is primarily due to the high rigidity and layered structure of MoS_2_ nanosheets, along with the interface optimization brought about by PDA modification. The increase in the rigidity and strength of the composite material is attributed to the high elastic modulus of MoS_2_ and its stress dispersion and load-bearing role within the composite structure. However, the dispersion and interface compatibility of MoS_2_ nanosheets in the PI matrix are limiting factors for their mechanical reinforcement effect. Building on this, the introduction of PDA not only improves the dispersion of the nanosheets but also forms cross-linked structures with active groups on their surface and the PI matrix [38]. This enhances the bonding between the nanosheets and the film, thereby improving the overall mechanical stability and durability of the composite material.

### 3.4. Dielectric Properties

Figure 7 shows the variation in the dielectric constant as well as the dielectric loss angle tangent with frequency for pure PI, MoS_2_/PI, and MoS_2_@PDA/PI. From the figure, it can be seen that the dielectric constant of MoS_2_/PI is between 4.5 and 4, while the dielectric constant of pure PI is between 3 and 3.5. This result indicates that the doping of nano-MoS_2_ can significantly enhance the dielectric constant of PI films. The dielectric constant of MoS_2_@PDA/PI, on the other hand, is between 3.5 and 4. It is speculated that the PDA parcel improves the interfacial bonding and compatibility between the nano-MoS_2_ and the PI base material [39]. This facilitates the polarization of the dipole under the electric field and thus reduces the permittivity of the PI. Similar to the dielectric constant, the dielectric loss angle tangent is the largest for MoS_2_/PI, followed by MoS_2_@PDA/PI, and the smallest for pure PI. The incorporation of MoS_2_ nanoparticles increases the polarization loss of the film, resulting in an increase in the angular value of the dielectric loss of MoS_2_/PI [40], whereas the wrapping of PDA effectively reduces the polarization loss of the film. Thus, the angular value of dielectric loss of MoS_2_@PDA/PI was reduced. This suggests that the dielectric constant and dielectric loss angle tangent of the composite film can be effectively reduced through PDA encapsulation and the composite films with superior electrical properties can be obtained. In conclusion, the overall dielectric properties of PDA-coated MoS_2_-doped PI composites are better than those of MoS_2_/PI.

### 3.5. Electrical and Aging Characteristics

In order to investigate the breakdown and aging characteristics of composite films, we built a corresponding experimental test platform. As shown in Figure 8, the whole testbed is based on a permanent magnet direct-drive wind turbine with a capacity of 2.5 MW. During the experiment, the permanent magnet rotor of the generator is removed and only the stator part is retained. The stator part is connected to the PWM inverter via a cable of about 100 m in length. To facilitate measurement, the main insulation on the winding coils was removed before the experiment so that the first-turn conductors of the coils of each group in a generator stator winding branch in phase U were exposed. This was used as a measurement point for the voltage signals from the coils to the ground and between the coils. In addition, the insulation on the nose of the first coil near the motor terminals is removed so that the conductor of each turn of the coil is exposed as a measurement point for the inter-turn voltage signal. During measurement, the coils are numbered 1, 2, …, and 8 in order from terminal to neutral. The high-voltage probe used for the measurement is the P6015A with a bandwidth of 75 MHz. During the experiment the gate voltage was set to 1.12 kV, the carrier frequency was 3 kHz, and the fundamental frequency was 35.7 Hz.

Figure 9 shows the Weibull distribution plots of breakdown voltage and partial discharge onset voltage for pure PI, MoS_2_/PI, and MoS_2_@PDA/PI. As can be seen from the figure, the Weibull distribution curve of the breakdown voltage of MoS_2_/PI is located on the left side of pure PI and the position is very close. When the breakdown probability is 63.2%, the breakdown voltages of MoS_2_/PI and pure PI films are basically the same: 17.92 kV and 19.56 kV, respectively.

The former is slightly weaker than the latter by about 9%. The Weibull distribution curve of the breakdown voltage of MoS_2_@PDA/PI is located on the right side of pure PI and they are farther away from each other. The breakdown voltage of PDA is 23.99 kV at 63.2% breakdown probability which is 1.23 times higher than pure PI film; this comes down to the semiconductor characteristic of MoS_2_ [41]. Under the action of a strong electric field, the electrons in the matrix are easily excited from the valence band to the conduction band and become free electrons, leading to a decrease in breakdown voltage. However, the PDA coating on the surface of MoS_2_ changes the energy level of the MoS_2_ surface [42]. It forms a kind of energy barrier that hinders the excitation and migration of electrons and reduces the number of free electrons, thus effectively enhancing the breakdown voltage of the composite film. Contrary to the internal breakdown voltage, the partial discharge initiation voltage of pure PI, MoS_2_/PI, and MoS_2_@PDA/PI increases sequentially. Specifically, for different types of nanocomposite films, the breakdown voltages of pure PI, MoS_2_/PI, and MoS_2_@PDA/PI with partial discharge probability of 63.2% are 2.48 kV, 2.68 kV, and 3.08 kV, respectively. MoS_2_/PI and MoS_2_@PDA/PI are enhanced by about 8% and 24.1%, respectively, compared with pure PI films. The reason is that MoS_2_ can improve the mechanical flexibility of PI to reduce the air gap and thus enhance the corona voltage, while the semiconductor property of nano-MoS_2_ will reduce the corona voltage [43]. The interaction between the two leads to a slightly stronger partial discharge onset voltage for MoS_2_/PI than for pure PI. Meanwhile, the phenolic hydroxyl and amine groups in the PDA molecular structure can form stable chemical bonds and physical adsorption with the surfaces of many materials [44]. This structure is conducive to further enhancing the interfacial bonding between the composite material and the metal winding, and this tight bonding reduces the presence of air gaps and the risk of air gap discharge. Therefore, the starting discharge voltage of MoS_2_@PDA/PI is enhanced due to the above reasons. In summary, MoS_2_@PDA/PI has excellent electrical insulation properties compared to pure PI and MoS_2_/PI, and the breakdown voltage of pure PI is higher than that of MoS_2_/PI, while the partial discharge onset voltage is lower than that of MoS_2_/PI.

The insulation life of pure PI, MoS_2_/PI, and MoS_2_@PDA/PI at different aging voltages is shown in Figure 10. From the figure, it can be seen that the insulation life of MoS_2_/PI is weaker than that of pure PI at all voltage levels. Under the aging voltage of 9 kV, the insulation life of MoS_2_/PI and pure PI are 224 h and 258 h, respectively. At 13 kV, the insulation life of MoS_2_/PI and pure PI is 9.68 h and 11.13 h, respectively. Overall, the insulation life of pure PI is about 15% higher than that of MoS_2_/PI. For MoS_2_@PDA/PI, the insulation life is greater than that of pure PI and MoS_2_/PI films. Its insulation life reaches 330 h at an aging voltage of 9 kV which is about 57.3% and 27.9% longer than MoS_2_/PI and pure PI, respectively. The insulation life is 14.36 h at 13 kV and is about 48.3% and 29% higher than MoS_2_/PI and pure PI. The main reason for this phenomenon is that pure MoS_2_ and PDA-coated MoS_2_ have different effects on the generation and movement of electrons in the polymer matrix. In the meantime, it has different effects on the destruction process of polymer molecular chains. In addition, the capture of electrons and holes in the trap may emit high-energy UV light [45,46,47]. High-energy UV light can also cause polymer chains to break. Polymer molecular chain breakage creates partial low-density regions that increase the inhomogeneity of the material; this is beneficial to the formation of discharge channels and makes the material less insulating. Thus, the lifetime of MoS_2_/PI is usually shorter. When PDA is coated on the surface of MoS_2_, it effectively changes the electronic properties of MoS_2_ and reduces its electrical conductivity in PI materials [48,49]. This change can improve the life of the composite film. In summary, the aging insulation life of MoS_2_@PDA/PI is significantly stronger than that of PI film, while PI film is stronger than MoS_2_/PI.

## 4. Conclusions

In this research, a MoS_2_@PDA/PI composite film was created by adding MoS_2_ nanoparticles coated with PDA into a PI matrix. In terms of mechanical properties, the friction coefficient and wear rate of the MoS_2_@PDA/PI composite showed enhancements of 38.4% and 27.8%, respectively, compared to pure PI. In terms of tensile properties, there was an increase of 44.8% in tensile strength, 25.6% in elastic modulus, and 55.3% in elongation at break, relative to pure PI.

In terms of electrical insulation properties, the breakdown voltage and the partial discharge inception voltage for MoS_2_@PDA/PI film increased by 22.6% and 24.1%, respectively, compared to pure PI. The long-term electrical aging tests revealed that compared to pure PI, a significant increase of 27.9% in electrical aging lifetime was exhibited by the MoS_2_@PDA/PI film, attributable to its superior mechanical, tribological, and insulation properties. These results provide critical technical support for enhancing the reliability of associated electrical equipment.

## Figures and Tables

**Figure 1 polymers-16-00546-f001:**
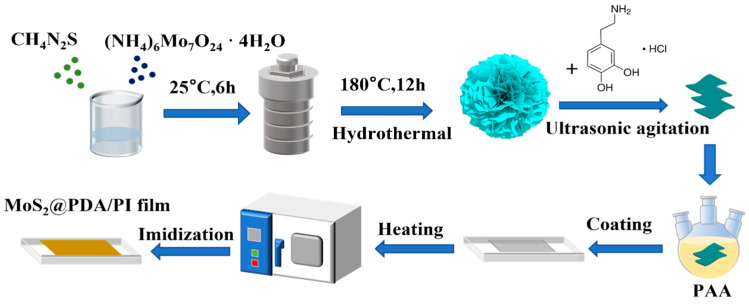
Schematic diagram of the synthesis process of MoS_2_@PDA/PI composites.

**Figure 2 polymers-16-00546-f002:**
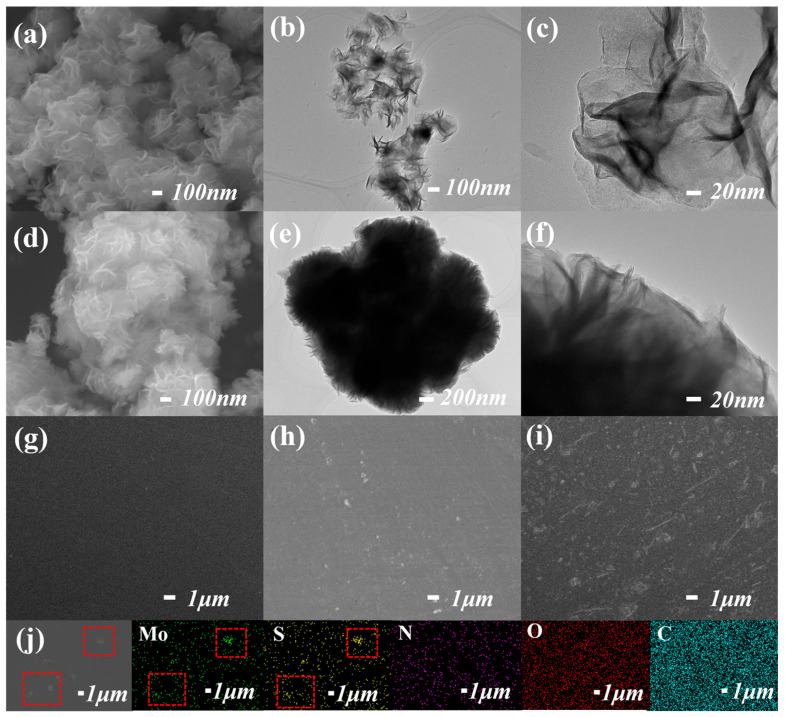
SEM (**a**) and TEM (**b**,**c**) images of MoS2; SEM (**d**) and TEM (**e**,**f**) images of MoS2@PDA; (**g**–**i**) are the SEM images of PI, MoS2/PI, and MoS2@PDA/PI; (**j**) is the SEM image and element mapping of MoS_2_@PDA/PI. The red dotted frames highlight the distribution locations of MoS_2_@PDA nanosheets.

**Figure 3 polymers-16-00546-f003:**
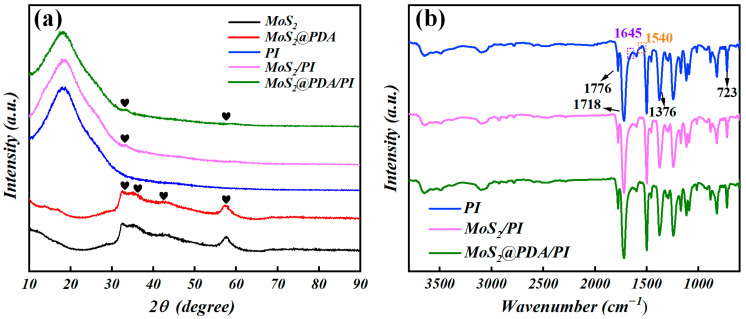
(**a**) The XRD pattern of MoS_2_, MoS_2_@PDA, PI, MoS_2_/PI, and MoS_2_@PDA/PI; (**b**) FT-IR spectrum of the PI, MoS_2_/PI, and MoS_2_@PDA/PI. The heart symbol represents the XRD diffraction peaks.

**Figure 4 polymers-16-00546-f004:**
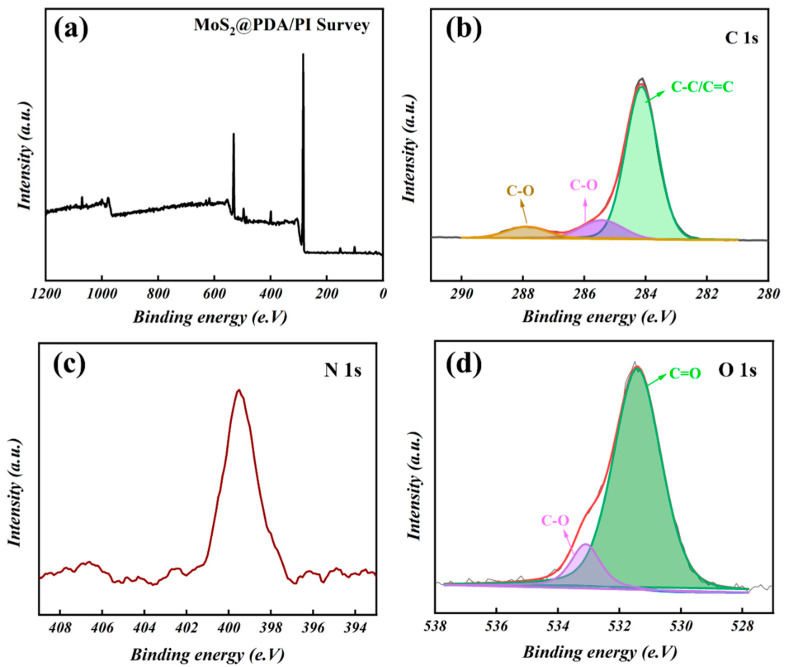
(**a**) The full spectra, (**b**) C 1s, (**c**) N 1s, and (**d**) O 1s of MoS_2_@PDA/PI.

**Figure 5 polymers-16-00546-f005:**
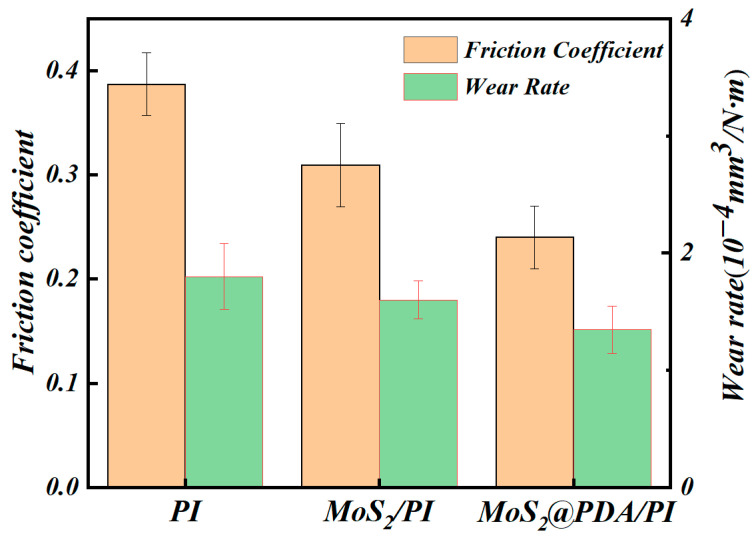
The friction coefficient and wear rate of PI, MoS_2_/PI, and MoS_2_@PDA/PI.

**Figure 6 polymers-16-00546-f006:**
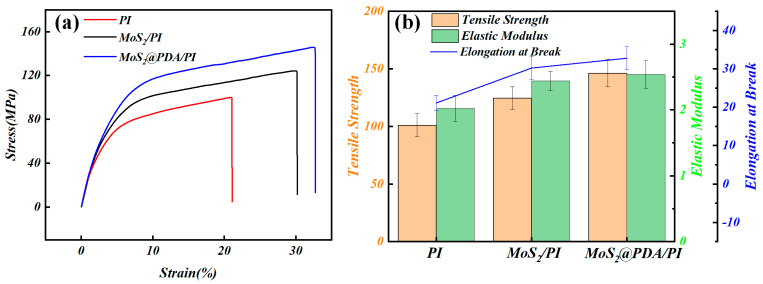
(**a**) The strain–stress curve of pure PI, MoS_2_/PI, and MoS_2_@PDA/PI composites; (**b**) tensile strength, elastic modulus, and break elongation of pure PI, MoS_2_/PI, and MoS_2_@PDA/PI composites.

**Figure 7 polymers-16-00546-f007:**
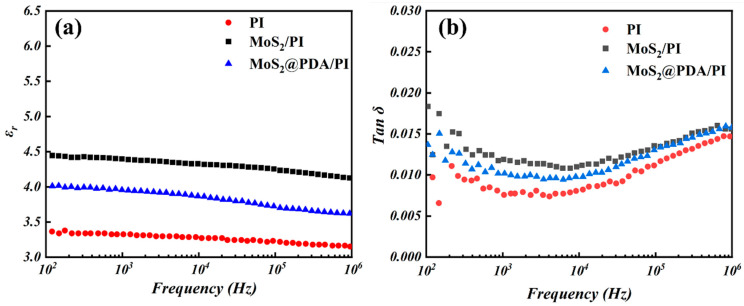
Dielectric properties of pure PI, MoS_2_/PI, and MoS_2_@PDA/PI: (**a**) dielectric constant and (**b**) dielectric loss tangent.

**Figure 8 polymers-16-00546-f008:**
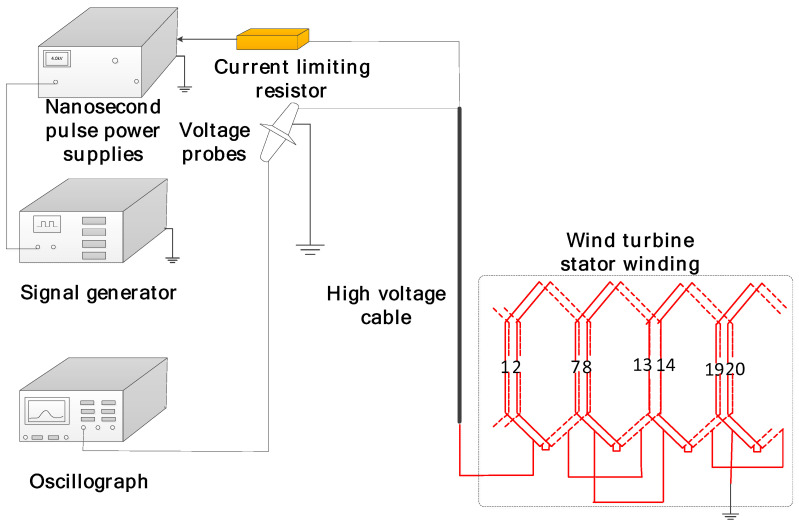
Structure of the pulse aging experimental device.

**Figure 9 polymers-16-00546-f009:**
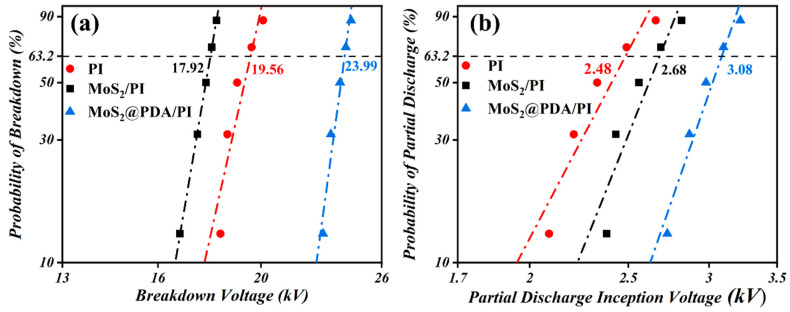
(**a**) Breakdown voltage and (**b**) partial discharge inception voltage of pure PI, MoS_2_/PI, and MoS_2_@PDA/PI.

**Figure 10 polymers-16-00546-f010:**
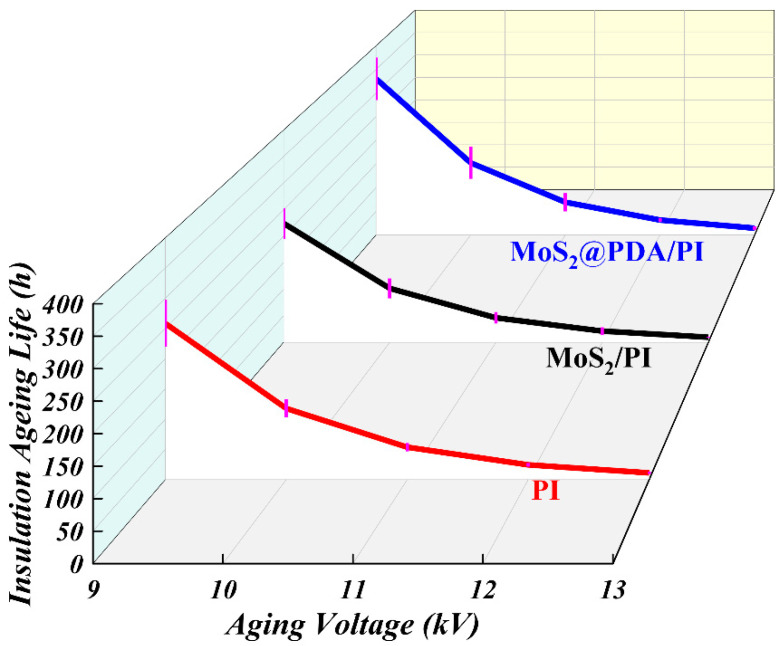
Insulation life of pure PI, MoS_2_/PI, and MoS_2_@PDA/PI under different aging voltages.

## Data Availability

Data are contained within the article.

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
