# Peer review of "Preparation of MoS2@PDA-Modified Polyimide Films with High Mechanical Performance and Improved Electrical Insulation"

_polymers, 2024, doi:10.3390/polym16040546_

Round 1

Reviewer 1 Report

Comments and Suggestions for Authors

The manuscript reported the composite formation of MoS2 and polydopamine to advance the insulation properties of polyimide (PI). The synthesized materials were characterized using various techniques such as FE-SEM, TEM, FT-IR, XRD, and XPS to confirm the morphology and composition of the final products. The MoS2@PDA/PI composite shows good insulation performance better than the pure PI. The manuscript is good and shows good data with well presentation.

Please address the following comments:

A)   In the experimental part: 1- The titles of 2.2 and 2.3 are the same. Please revise this section. 2- The 2.4 section needs more revision, it is highly recommended to revise this section carefully. 3-

B)   Results and discussion:

1-    It is highly recommended to add the EDX-SEM mapping to confirm the atomic distribution and the composition of the prepared materials.

2-    The conclusion part is too long, it is better to reduce the conclusion part by focusing on the merit of the figure.

Comments on the Quality of English Language

The English language is acceptable

Author Response

Dear Reviewer,

Thank you for your insightful comments and suggestions regarding our manuscript titled "Preparation the High Mechanical Performance of MoS2@PDA Modified Polyimide Film for Enhanced Electrical Insulation". We have thoroughly reviewed your feedback and have made corresponding revisions to our manuscript.

For your convenience, we have detailed the responses to your comments in the attached document, where each point of feedback is addressed specifically. Additionally, we have included the revised version of our manuscript for your review.

We believe these revisions have significantly improved our manuscript and hope that it now meets the standards for publication in Polymers. Thank you once again for your valuable input and guidance.

Reviewer 2 Report

Comments and Suggestions for Authors

Dear Professor, Editor of Polymers,

Thank you very much for invite me to review the manuscript under the title of “Preparation the High Mechanical Performance of MoS2@PDA Modified Polyimide Film for Enhanced Electrical Insulation

I studied this manuscript very well. This manuscript showed good results and can be accepted for publication after the authors modified the following;

  1. The authors should avoid the long sentence in the whole manuscript, especially in the abstract and the introduction sections.
  2. In page 4, the authors should add a suitable reference to support this sentence “TEM image (2c), the color of MoS2 single layer is lighter, indicating that the thickness of the nanosheet is thin.”   
  3. In page 4, the authors stated that “Peaks at 723 cm-1 could be ascribed to the C=O bond in polyimide”, I don’t think that this band is related to C=O bond, please check that or add a suitable reference to support that.
  4. In page 5, the authors should add a suitable reference to support this sentence “The N orbital shifts towards low energy compared to pure PI films in the literature, which is due to the non-covalent interaction between PI and MoS2@PDA”
  5. In page 6, the authors should rewrite these sentences “These figures show a significant decrease of 38.4% and 27.8%, respectively, compared to the pure PI film.” and “The result is a more uniform dispersion of MoS2@PDA nanosheets in the PI matrix, which further reduces both the wear rate and friction coefficient, achieving enhanced resistance to mechanical wear”
  6. In sections 3.3., 3.4., 3.5., the authors write the mechanical properties, please correct this mistake.
  7. In page 8, 3.4. should be 2.5.

Comments on the Quality of English Language
  1. The authors should avoid the long sentence in the whole manuscript, especially in the abstract and the introduction sections.

Author Response

(The authors gave the same response as above.)
